# The Self-Expandable Impella CP (ECP) as a Mechanical Resuscitation Device

**DOI:** 10.3390/bioengineering11050456

**Published:** 2024-05-03

**Authors:** Sebastian Billig, Rachad Zayat, Siarhei Yelenski, Christoph Nix, Eveline Bennek-Schoepping, Nadine Hochhausen, Matthias Derwall

**Affiliations:** 1Department of Anesthesiology, Faculty of Medicine, RWTH Aachen University, Pauwelsstraße 30, 52074 Aachen, Germany; 2Department of Cardiothoracic Surgery, Heart Center Trier, Barmherzigen Brüder Hospital Trier, 54292 Trier, Germany; 3Department of Thoracic Surgery, Medical Faculty RWTH Aachen University, Pauwelsstrasse 30, 52074 Aachen, Germany; 4Abiomed Europe GmbH, 52074 Aachen, Germany; 5Department of Anesthesia, Critical Care and Pain Medicine, St. Johannes Hospital, 44137 Dortmund, Germany

**Keywords:** left ventricular assist device, mechanical circulatory support, cardiac arrest, cardiopulmonary resuscitation, swine model, Impella ECP

## Abstract

The survival rate of cardiac arrest (CA) can be improved by utilizing percutaneous left ventricular assist devices (pLVADs) instead of conventional chest compressions. However, existing pLVADs require complex fluoroscopy-guided placement along a guidewire and suffer from limited blood flow due to their cross-sectional area. The recently developed self-expandable Impella CP (ECP) pLVAD addresses these limitations by enabling guidewire-free placement and increasing the pump cross-sectional area. This study evaluates the feasibility of resuscitation using the Impella ECP in a swine CA model. Eleven anesthetized pigs (73.8 ± 1.7 kg) underwent electrically induced CA, were left untreated for 5 min and then received pLVAD insertion and activation. Vasopressors were administered and defibrillations were attempted. Five hours after the return of spontaneous circulation (ROSC), the pLVAD was removed, and animals were monitored for an additional hour. Hemodynamics were assessed and myocardial function was evaluated using echocardiography. Successful guidewire-free pLVAD placement was achieved in all animals. Resuscitation was successful in 75% of cases, with 3.5 ± 2.0 defibrillations and 1.8 ± 0.4 mg norepinephrine used per ROSC. Hemodynamics remained stable post-device removal, with no adverse effects or aortic valve damage observed. The Impella ECP facilitated rapid guidewire-free pLVAD placement in fibrillating hearts, enabling successful resuscitation. These findings support a broader clinical adoption of pLVADs, particularly the Impella ECP, for CA.

## 1. Introduction

Despite significant advances in cardiopulmonary resuscitation (CPR) over the past 60 years, cardiac arrest (CA) still results in unacceptably high morbidity and mortality [1,2,3]. High-quality chest compressions with minimal interruptions are critical to improve both survival and good outcomes of CA [4,5,6]. In line with this, previous research in a large animal model of CA has demonstrated that using a minimally invasive percutaneous left ventricular assist device (pLVAD) (Impella 2.5) instead of manual chest compressions can double survival rates and improve clinical outcomes compared to manual chest compressions [7]. A combination of chest compressions with a pLVAD for resuscitation has also shown promising results, with a good neurologic outcome in a preclinical animal model [8]. In addition, pLVADs have successfully been used in individual clinical cases for resuscitation [9].

Yet, the complex implantation process of the pLVAD prohibits its routine use during CA even in a catheter laboratory. Firstly, vascular access for the pLVAD has to be established either by a percutaneously inserted vascular sheath or, less frequently, by surgical techniques [10,11]. Next, a soft wire must be advanced via the aortic valve into the left ventricle, which is then exchanged to a stiffer guidewire. The pLVAD can finally be implanted along the guidewire using the Seldinger technique under fluoroscopic guidance—a time-consuming procedure. Furthermore, the placement of large-bore sheaths can lead to major complications such as bleeding, vascular damage or limb ischemia [12]. Nevertheless, clinical use of pLVADs during CA has already been described multiple times by other groups. In summary, these studies show great variances in survival (6–50%) and observed vascular complication rates (7–50%) [9,13,14,15].

This study explores whether the recently developed self-expandable Impella CP (ECP) pLVAD could offer a sufficient alternative to previous models in the context of resuscitation. The design of the Impella ECP as a self-expandable pLVAD allows for left ventricular insertion without the need for a previously placed guidewire and reduces the required introducer sheath size. Our study aims to determine whether sufficient mechanical resuscitation could be achieved in a large animal model of CA using the Impella ECP, thus paving the way for its use in a wider spectrum of clinical settings.

## 2. Materials and Methods

### 2.1. Impella ECP pLVAD

The Impella ECP device has not yet been approved for use in humans; various clinical trials are currently in progress (www.clinicaltrials.gov accessed on 3 April 2024; NCT05334783 and NCT04477603). We used two different versions of the Impella ECP. In the first phase of this study, we used the original ECP pump for three animals, while we used an adapted pump model in the subsequent experiments (compare Figure 1A,B). The adapted pump had a modified outflow cannula with an additional blood outlet to address the differences between the pig and the human anatomy.

### 2.2. Preparation

The reporting in the manuscript complies with the ARRIVE Guidelines 2.0 [16]. For an overview of the complete experimental procedure, please see Figure 1C.

Azaparone (6 mg/kg; Stresnil, Janssen-Cilag GmbH, Neuss, Germany) was intramuscularly injected followed by intravenous induction of narcosis by propofol (2 mg/kg; Propofol 1% MCT, Fresenius Kabi Austria GmbH, Graz, Austria) and fentanyl (5 µg/kg; Fentanyl-Jannsen, Janssen-Cilag GmbH). The animals were endotracheally intubated and mechanically ventilated (Cato, Dräger, Lübeck, Germany). The ventilator was set to an inspired oxygen fraction of 0.3, a tidal volume of 10 mL/kg and a positive end-expiratory pressure of 5 cmH_2_O. The respiratory frequency was adjusted to maintain a physiologic end-tidal carbon dioxide level from 35 to 40 mmHg. Narcosis was continued by intravenous propofol (5–10 mg/kg/h) and fentanyl (5 µg/kg/h) dosing. Ringer’s solution was administered at a rate of 4 mL/kg/h. ECG monitoring, pulse oximetry and cerebral oximetry (INVOS 5100c cerebral oximeter, Medtronic; INVOS cerebral oximetry adult sensors, Covidien, Dublin, Ireland) were performed. Body temperature was kept at 38.2 ± 0.2 °C using convective heating (Warm Touch 5300A, Covidien). Defibrillation electrodes (QUIK-COMBO, Physio-Control, Redmond, WA, USA) were applied to the shaved chest of the animals and attached to a defibrillator (Lifepak 12, Medtronic, Minneapolis, MN, USA). Under ultrasonic guidance introducer sheaths (9F percutaneous sheath introducer set, Arrow, Reading, PA, USA) were placed into the right jugular and the right femoral vein. A hexalumen Swan-Ganz catheter (744HF75, Edwards Lifesciences, Irvine, CA, USA) was flow-directed into the pulmonary artery and connected to a monitor (Vigilance VGS2, Edward Lifesciences, Irvine, CA, USA). For the measurement of arterial pressure and the collection of blood samples, a 4F arterial catheter (Arterial leadercath, Vygon, Ecquen, France) was placed in the right femoral artery. A 10F introducer sheath (10F Introducer Avanti, Cordis, Miami Lakes, FL, USA) was inserted into the left femoral artery. After successful catheter placement, the animals received 100 IU/kg heparin (B. Braun, Melsungen, Germany) intravenously.

Transesophageal echocardiography (TEE) was applied to visualize the pLVAD and the aortic valve during the implantation process [17]. Furthermore, we used a small surgical subxiphoid access to obtain apical views of both ventricles. TEE was conducted using a GE Vivid E9 system equipped with a 6VT-D (3.0–8.0 MHz, GE Vingmed Ultrasound AS, Horten, Norway) probe. For the subxiphoid access, a 4Vc-D (1.4–5.2 MHz, GE) transthoracic probe was used. Examinations were carried out by certified sonographers.

### 2.3. Cardiac Arrest and Resuscitation

After an equilibration period of 30 min, the animals received another dose of heparin (100 IU/kg). A 5-F pacing catheter was inserted into the right ventricle via jugular access. Ventricular fibrillation (VF) was induced electrically and confirmed by ECG and a non-pulsatile arterial blood pressure curve. Ventilation, narcotics and fluid administration were paused. CA was left untreated for 5 min until resuscitation was initiated; ventilation with 100% oxygen was resumed and Ringer’s solution was infused at a rate of 200 mL/min. The Impella ECP pLVAD was then implanted into the left ventricle via the previously placed femoral introducer sheath (Figure 2) and started simultaneously with the administration of 1 mg norepinephrine (Arterenol, Sanofi, Frankfurt, Germany). The required time for the implantation of the pLVAD was measured. Validation of the positioning of the device was conducted using fluoroscopy and TEE. After 2 min of CPR, a biphasic shock with an energy level of 360 J was applied. In case the shock proved unsuccessful, subsequent shocks were delivered every two minutes. If VF persisted following the second shock, 1 mg of norepinephrine was administered and defibrillations were repeated at two-minute intervals. Animals failing to achieve return of spontaneous circulation (ROSC)) within 10 min were classified as non-resuscitated.

### 2.4. Follow-Up

The animals were kept in narcosis for another six hours following ROSC to monitor cardiocirculatory and metabolic recovery. Fluid administration was limited to 4 mL/kg/h and F_i_O_2_ was reduced to 0.3. Five hours following ROSC, the pLVAD was weaned and removed. One hour after the removal, hemodynamics were evaluated without the support of the device. Animals were finally sacrificed.

### 2.5. Measurements

Hemodynamic data were measured by the indwelling catheters and recorded continuously (LabVIEW 2010, National Instruments, Austin, TX, USA). Coronary perfusion pressure (CPP) was estimated by subtracting the diastolic right atrial pressure from the diastolic femoral arterial pressure [18]. During resuscitation, we used the mean right atrial pressure and the mean femoral pressure due to the missing pulsatility in the blood pressure signal of the non-contracting ventricle. The measured cerebral oximetry values were normalized to the baseline (BL) values. Blood samples were collected at baseline, after 10 and 30 min and every full hour following ROSC (PR 10, PR 30 and PR60-PR360). Blood gas analysis was performed for arterial and mixed venous samples using a standard blood gas analyzer (ABL 700; Radiometer, Copenhagen, Denmark). Blood cell counting was carried out using an automated blood cell machine at baseline and 6 h following ROSC (Celltac-alpha VET MEK-6550K; Nihon Koden, Rosbach, Germany).

### 2.6. Statistical Analysis

Three animals that were resuscitated in the pilot phase of the experiments using the unmodified Impella ECP were not included in the analysis. Statistical evaluation was performed using SPSS (SPSS statistics Version 28, IBM, Armonk, NY, USA). Graphs were plotted using GraphPad PRISM 10 (GraphPad Software, San Diego, CA, USA). The data are presented as mean ± standard deviation (SD) unless indicated otherwise. Normal distribution of the data was checked by using diagnostic plots and the Kolmogorov–Smirnov test. Comparisons between different time points were examined using a repeated-measures analysis of variance (ANOVA) with Greenhouse–Geisser correction, followed by Tukey’s post hoc test or a *t*-test. In case normal distribution of the data was not present, the non-parametric Friedman’s test was used. The null hypothesis was rejected for *p*-values < 0.05.

## 3. Results

For pLVAD insertion, a regular 10F arterial introducer was placed successfully in all (n = 11) subjects. The dedicated 9F ECP introducer was not used because it was too long for the pig anatomy. No adverse events, such as hemorrhage, vascular dissection or limb ischemia attributable to vascular access, were documented in our observations. Implantation of the Impella ECP pLVAD into the fibrillating ventricle was achieved in all 11 animals without a guidewire under fluoroscopic control. Advancement into the ventricle was technically easy to perform, since the stiff shaft of the device allowed for good control during implantation. Gentle pressure at the catheter shaft was sufficient to maneuver the tip of the catheter across the aortic valve (Figure 2; Appendix A). The inlet of the pLVAD could then be positioned into the ventricle, the flexible outflow tube could be placed at the level of the aortic valve and the outlet could be positioned in the ascending aorta. In total, 59 ± 8 s were required to place the Impella ECP in the final left ventricular position via the previously placed femoral arterial introducer sheath. In case it would be necessary, the handle of the device enables to us rotate the pLVAD along the longitudinal axis (Appendix A).

Resuscitation was not successful in two out of three animals using the non-modified Impella ECP during the first phase of the study. The subsequent experiments with the adapted Impella ECP resulted in successful resuscitation in six out of eight (75%) animals (Table 1). One of those animals could not be resuscitated due to failure to achieve ROSC within the 10 min of CPR after 5 min of untreated CA. In the other animal, the Impella ECP’s drive shaft was damaged during the passage of the introducer. The damage resulted in a device malfunction, causing inability to achieve a sufficient pump flow. A total of 3.5 ± 2.0 defibrillations and 1.8 ± 0.4 mg norepinephrine were required until ROSC could be achieved in each animal. All animals that achieved ROSC survived the 6 h follow-up period.

Figure 3 visualizes the main hemodynamic parameters during untreated CA and after the start of the resuscitation by using the pLVAD. The onset of VF immediately led to a massive drop in mean arterial pressure (MAP), the calculated CPP and relative cerebral oxygenation (relS_c_O_2_), while the CVP increased. After the pLVAD was started, MAP and the calculated CPP increased immediately. The relS_c_O_2_ did not increase during the first two minutes of resuscitation. The calculated CPP one minute after the device was started was 14.3 ± 4.4 mmHg (CPP before vs. CPP 1 min after the device was started, *p* < 0.001). A total of 3 min after the device was started, the CPP had increased to 20.3 ± 6.4 mmHg (CPP 1 vs. 3 min after the device was started, *p* = 0.03). After ROSC, the resuscitated animals showed hypertensive blood pressure values, followed by a period of mild arterial hypotension (PR 30: MAP = 57 ± 20 mmHg) that normalized over the next hours (Table 2). Hemodynamic stabilization was achieved in all animals without the application of catecholamines after ROSC. Due to the global ischemia–reperfusion damage, the animals developed lactatemia which peaked 30 min post-ROSC at 8.9 ± 1.7 mmol/L (Table 2). Three hours after ROSC, lactate returned to baseline values (*p* > 0.99). Heart rate (HR), mean arterial pressure (MAP), mean pulmonary arterial pressure (MPAP), central venous pressure (CVP) and pulmonary artery wedge pressure (PCWP) 6 h after ROSC showed no significant differences compared to baseline (HR, *p* = 0.98; MAP, *p* = 0.24; MPAP, *p* = 0.68; CVP, *p* = 0.12; PCWP, *p* = 0.94). However, there was a trend toward a lower cardiac output (CO) (6.9 ± 1.1 L/min vs. 5.1 ± 0.9 L/min, *p* = 0.06) 6 h after ROSC compared to baseline. Hemoglobin (Hb) was stable (*p* = 0.45), while platelets showed a statistically insignificant decrease 6 h after ROSC (280 ± 65/nl vs. 204 ± 37/nL, *p* = 0.09). 

Echocardiography revealed no pre-existing cardiac conditions or valve abnormalities in the studied subjects. Baseline left and right ventricular function was within normal limits (Table 3). A total of 30 min after resuscitation, a temporary severe restriction in left ventricular (LV) and right ventricular (RV) systolic function was observed, indicated by a decrease in LV and RV global longitudinal strain (GLS) (BL vs. PR 30 LV-GLS, *p* = 0.001; BL vs. RV-GLS, *p* = 0.002). However, 6 h after ROSC, LV function was comparable to baseline values (LV-EF, *p* = 0.99; LV-GLS, *p* = 0.65). RV function did recover within the follow-up but was still restricted after pLVAD removal (BL vs. 6 h RV-GLS −26.2 ± 4.4% vs. −18.6 ± 6.3%; *p* = 0.08). The parameters of diastolic dysfunction showed no significant changes after the removal of the device compared to the BL values. Significant aortic valve insufficiency could be excluded in all animals after device removal. 

## 4. Discussion

In this exploratory study, the new Impella ECP pLVAD revealed excellent properties for emergency use during CA. The present study confirmed the device’s suitability for implantation during CA through a straightforward, guidewire-free procedure. Echocardiographic assessments verified the integrity of the aortic valve following retrograde insertion of the pLVAD. Furthermore, the Impella ECP provided effective hemodynamic support in a large animal model of CA, resulting in a high rate of ROSC post-defibrillation.

Preclinical [7,8] and clinical [9,13,14] data suggest that the implantation of a transvalvular pLVAD during CA is feasible and can lead to good neurologic outcomes. Particularly in the catheter laboratory setting, pLVADs can provide rapid and uninterrupted circulatory support while the underlying cause of CA can be solved [8,19]. To facilitate further development of this technique, a safe and fast implantation technique of the device that fits the urgency of CA is essential. The standard device implantation procedure involves an exchange of a guidewire after initial placement of a guiding catheter (i.e., pigtail catheter) in the LV. Finally, the pLVAD can be introduced into the LV using the mono-rail technique. For the insertion during CA, thoracic compressions are typically used to open the aortic valve [13] and direct the tip of the wire across the valve. In our previous trials, the guidewire was placed in the LV before induction of CA for this reason [7,20]. Successful deployment of the Impella ECP via a previously established introducer sheath could be achieved in less than a minute on average during this trial. Minimizing low- and no-flow times during CA is crucial to reduce ischemia, emphasizing the potential importance of a rapidly implantable continuous mechanical circulatory support device. The need for expedited and simplified pLVAD implantation extends beyond CA scenarios, benefiting patients undergoing treatment for cardiogenic shock or protected percutaneous coronary interventions. The described implantation technique without a guidewire may mitigate potential arrhythmias or perforations induced by guidewires. While fluoroscopic visualization of the guidewires and the Impella pLVAD during implantation is recommended by the manufacturer, successful Impella placement without fluoroscopy, controlled by TEE only, has already been described previously [21,22]. The simplified implantation technique described here for the Impella ECP could lead to further spread of the TEE-only controlled implantation technique, allowing for on-site implantation outside a catheter laboratory. Insertion of the Impella ECP without the use of fluoroscopy was not studied in this trial; however, the simple insertion of the device suggests that it is likely to be successful (Appendix A).

Vascular access poses a critical challenge for transvalvular pLVAD implantation, especially in patients with peripheral arterial occlusive disease [23]. Furthermore, the likelihood of vascular complications like bleeding, vascular injury or limb ischemia increases with the size of the vascular sheaths [24,25,26]. However, a reduction in the introducer sheath size jeopardizes the high flow rate of the implanted pLVAD, which is mandatory for sufficient hemodynamic support. The self-expanding pump housing of the Impella ECP can overcome this interdependence. It enables both a small introducer sheath size and an adequate pump flow. In this trial, the Impella ECP pLVAD was implanted via a 10F introducer sheath while it finally expanded to a diameter of 21F, delivering up to 3.9 L/min.

During CA, the Impella ECP pLVAD effectively generated perfusion, as evidenced by the increase in MAP and the calculated CPP, resulting in a 75% ROSC rate after defibrillation attempts. The calculated CPP of 20.3 ± 6.4 mmHg three minutes after the device was started was comparable to previous studies [7] of our group and a similar trial of Lotun et al., which reported a CPP of 21.2 ± 16.7 mmHg using an Impella 2.5 device [8]. The mean pump flow before defibrillation was 2.2 ± 0.3 L/min, which corresponds to approximately 33% of the animals’ native cardiac output of 6.8 L/min. In a previous study of our group using the Impella CP during resuscitation of approximately 35 kg weighing swine, the mean pump flow was 1.36 ± 0.02 L/min [7]. In comparison, manual chest compressions can only generate approximately 20% of the native cardiac output [27]. The ROSC rate of 75% in this small trial was satisfactory, although below the results of 100% from our previous trials [7,28]. Overall, we conclude that sufficient resuscitation could be achieved using the Impella ECP. Echocardiographic examinations indicated a transient restriction in LV and RV function 30 min post-ROSC. During the follow-up period, however, we observed an almost complete recovery of LV cardiac function, while RV cardiac function did not recover as significantly. Sufficient aortic valve function after pLVAD removal was confirmed and no valve insufficiency was observed. 

## 5. Limitations

The following limitations must be recognized when interpreting our results: First, this study was conducted in young and healthy animals without any pre-existing cardiac conditions. The animals did not have any aortic valve disease that might have complicated the insertion of the device. For catheterization of patients with aortic stenosis, it is known that retrograde access to the left ventricle is not practicable in 5% of the cases [29]. A pre-damaged heart (e.g., an ischemic cardiomyopathy) would moreover probably have shown a lower rate of ROSC as well as a more severe restriction in ventricular function after ROSC. The latter would lead to a more significant impact of the Impella support after ROSC. Second, the influence of the modification made to the blood outlet of the Impella ECP was not studied. Due to the pig anatomy, the outflow openings of an Impella ECP will reside in the descending aorta. The acute angle between the ascending and descending aorta may reduce flow through the outflow cannula. The modifications to the outflow cannula aimed to improve blood flow to the ascending aorta. Whether we achieved this goal or not cannot be proven by the current experimental set-up as it was not designed for this purpose. Furthermore, our study cannot quantify the important neurological outcome of the resuscitation with the Impella ECP. However, the focus of this feasibility study was the insertion process of the device and hemodynamics. From our point of view, a weaning of the animals from the ventilator for neurological assessment would not have been ethically justified during this proof-of-concept trial. Finally, our study did not include a reference group of animals treated by another pLVAD device or manual chest compressions. However, this pilot study was only designed to investigate the general feasibility of resuscitation by the Impella ECP with a focus on the implantation process.

## 6. Conclusions

Guidewire-free device placement is feasible and simple to perform with this novel Impella ECP device. Resuscitation in a large animal model of CA using the Impella ECP generated a favorable outcome in terms of hemodynamics. This study is the first step in the investigation of the Impella ECP pLVAD for resuscitation and further studies are necessary before clinical deployment can be considered. At the moment, the optimal setting for an Impella ECP-based resuscitation (e.g., catheterization laboratory, intensive care unit or out-of-hospital cardiac arrest) cannot be defined.

## Figures and Tables

**Figure 1 bioengineering-11-00456-f001:**
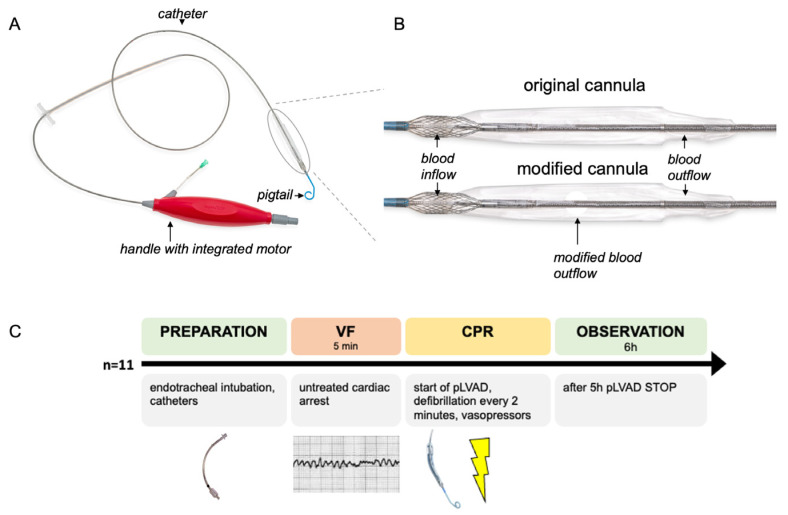
An illustration of the Impella ECP device and the experimental procedure. (**A**) A photograph of the Impella ECP: the device consists of an extracorporeal handle with the integrated motor and the catheter itself. The cannula with the blood in- and outflow is mounted at the distal part of the catheter. A pigtail at the distal part of the catheter ensures the stable position of the device in the left ventricle and avoids suction. (**B**) During the pilot phase of the experiment, the catheter was modified by adding an additional blood outflow window. This picture is showing the original cannula and the modified cannula. (**C**) This flowchart illustrates the individual steps of the experiment that were carried out in *n* = 11 female swine. CPR: cardiopulmonary resuscitation, pLVAD: percutaneous left ventricular assist device and VF: ventricular fibrillation.

**Figure 2 bioengineering-11-00456-f002:**
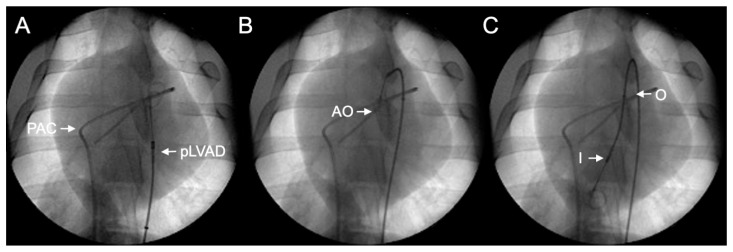
An illustration of the wireless Impella ECP implantation procedure using fluoroscopy. The Impella ECP percutaneous left ventricular assist device (pLVAD) (Abiomed Europe GmbH, Aachen, Germany) is placed into the left ventricle during ventricular fibrillation without the use of a guidewire. (**A**) The Impella ECP pLVAD is advanced into the descending thoracic aorta. PAC: pulmonary artery catheter; pLVAD: percutaneous left ventricular assist device. (**B**) The Impella ECP is further advanced; the distal pigtail is located at the level of the aortic valve. AO: approximate location of the aortic valve. (**C**) The final position of the device in the left ventricle. I: inlet; O: outlet.

**Figure 3 bioengineering-11-00456-f003:**
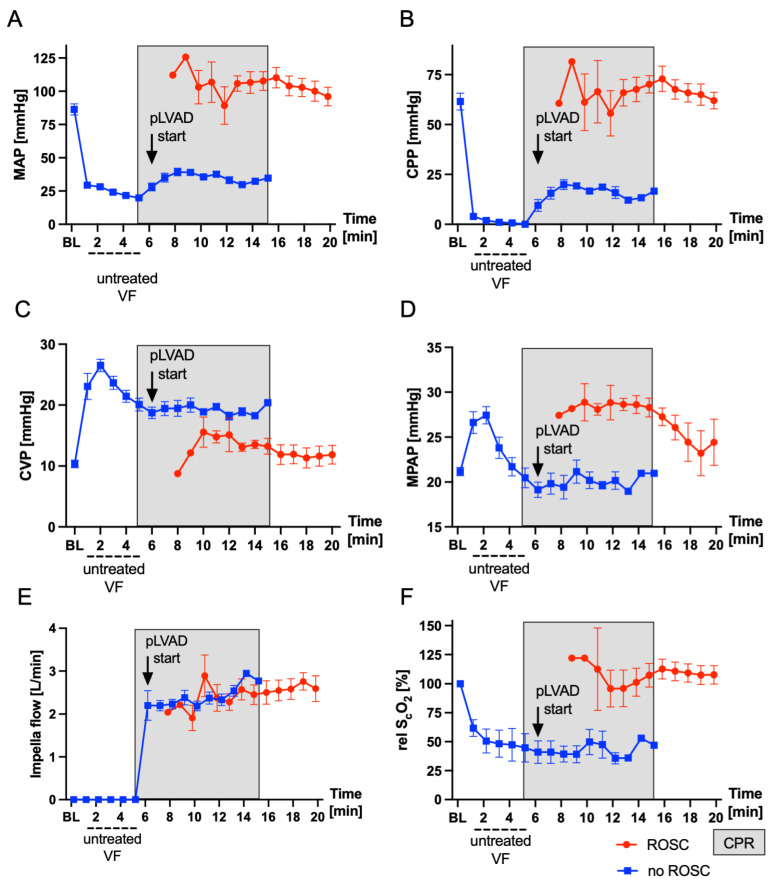
Hemodynamic parameters during cardiac arrest and CPR of n = 8 swine that were resuscitated using a modified Impella ECP pLVAD. (**A**) Mean arterial pressure. (**B**) The calculated cardiac perfusion pressure. (**C**) The central venous pressure. (**D**) Mean pulmonary arterial pressure. (**E**) The calculated Impella flow. (**F**) Relative cerebral oxygenation. The data are presented as the mean ± standard error of the mean (SEM). BL: baseline, CPP: cardiac perfusion pressure, CPR: cardiopulmonary resuscitation, CVP: central venous pressure, MAP: mean arterial pressure, MPAP: mean pulmonary arterial pressure, pLVAD: percutaneous left ventricular assist device, relS_c_O_2:_ relative cerebral oxygenation and VF: ventricular fibrillation. The red line represents the subjects with ROSC, while the blue line represents all subjects during untreated VF and the subjects with VF under ongoing CPR. The gray box indicates the resuscitation period (mechanical ventilation, fluid administration and vasopressors). The arrow indicates the start of the pLVAD.

**Table 1 bioengineering-11-00456-t001:** Resuscitation parameters.

Parameter	Value ± SD
Vascular access complications [n]	0
pLVAD implantation time [s]	59 ± 28
Implantation success [%]	100
Defibrillation attempts [n]	3.5 ± 2
Time to ROSC [min]	11.33 ± 2.07
Norepinephrine [mg]	1.8 ± 0.4
Survival [n] [%]	6/8; 75%

Implantation time, complications, survival rates, required vasopressors and defibrillation attempts in n = 8 swine that were resuscitated using a modified Impella ECP pLVAD. The data are presented as mean ± standard deviation. pLVAD: percutaneous resuscitation device; ROSC: return of spontaneous circulation.

**Table 2 bioengineering-11-00456-t002:** Hemodynamic parameters, blood gas data and cell count data.

	BL (n = 8)	PR 10 (n = 6)	PR30 (n = 6)	PR 120 (n = 6)	PR 300 (n = 6)	PR 360 (n = 6)
HR [bpm]	73 ± 15	144 ± 33	134 ± 43	86 ± 10	67 ± 11	70 ± 17
MAP [mmHg]	86 ± 12	94 ± 19	57 ± 20	77 ± 15	80 ± 9	79 ± 8
MPAP [mmHg]	21 ± 1	23 ± 5	21 ± 4	21 ± 3	23 ± 2	22 ± 2
CVP [mmHg]	10 ± 2	11 ± 2	12 ± 4	11 ± 2	12 ± 2	11 ± 2
PCWP [mmHg]	14 ± 3	15 ± 3	15 ± 2	15 ± 2	14 ± 2	13 ± 2
CO [L/min]	6.9 ± 1.1	10.0 ± 1.6	7.8 ± 2.0	6.1 ± 0.7	5.1 ± 0.4	5.1 ± 0.9
pLVAD flow [L/min]	0 ± 0	2.9 ± 0.5	3.3 ± 0.9	3.0 ± 0.8	3.0 ± 0.6	0 ± 0
p_a_O_2_ [mmHg]	144 ± 20	472 ± 94	173 ± 38	161 ± 10	167 ± 12	171 ± 14
p_a_CO_2_ [mmHg]	38 ± 2	42 ± 3	40 ± 4	37 ± 2	37 ± 1	38 ± 1
S_v_O_2_ [%]	62 ± 8	82 ± 4	53 ± 16	53 ± 9	52 ± 6	54 ± 9
pH	7.47 ± 0.04	7.3 ± 0.05	7.34 ± 0.02	7.45 ± 0.04	7.49 ± 0.00	7.49 ± 0.01
Hb [g/dL]	9.3 ± 0.7	11.4 ± 0.7	10.1 ± 0.9	9.5 ± 0.6	8.9 ± 0.9	8.8 ± 0.9
Lactate [mmol/L]	1.7 ± 2.3	8.1 ± 2.4	8.9 ± 1.7	4.8 ± 1.2	0.8 ± 0.1	0.8 ± 0.1
Glucose [mg/dL]	111 ± 20	185 ± 56	151 ± 71	129 ± 29	111 ± 12	104 ± 10
PLT/nL [n]	280 ± 65					204 ± 37
WBC/nL [n]	18.9 ± 4.4					13.3 ± 5.3

Hemodynamic parameters, blood gas data and cell count data of n = 8 swine that were resuscitated using a modified Impella ECP. The values are shown for baseline (BL) and 10 (PR10), 30 (PR30), 120 (PR120), 300 (PR 300) and 360 (PR360) minutes following the return of spontaneous circulation. CO: cardiac output, CVP: central venous pressure, Hb: hemoglobin concentration, HR: heart rate, MAP: mean arterial pressure, MPAP: mean pulmonary artery pressure, PCWP: pulmonary capillary wedge pressure, PLT: platelet count, pLVAD: percutaneous left ventricular assist device, p_a_CO_2:_ arterial carbon dioxide tension, p_a_O_2:_ arterial oxygen tension, S_v_O_2:_ mixed venous oxygenation and WBC: white blood cell count. The data are presented as the mean ± standard deviation (SD).

**Table 3 bioengineering-11-00456-t003:** Echocardiographic left and right ventricular function.

	BL (n = 7)	PR 30 (n = 5)	PR 300 (n = 5)	PR 360 (n = 5)
LV-EF [%]	59 ± 7	45 ± 9	66 ± 12	59 ± 19
LV-GLS [%]	−24.2 ± 3.3	−11.9 ± 5.8	−23.4 ± 3.8	−20.1 ± 4.1
RV-GLS [%]	−26.2 ± 4.4	−14.0 ± 5.9	−23.7 ± 3.2	−18.6 ± 6.2
RVD basal [mm]	28 ± 5	30 ± 7	30 ± 6	29 ± 4
TASV [cm/s]	10.8 ± 2.2	7.0 ± 3.3	8.4 ± 1.5	9.4 ± 1.5
E/A	1.8 ± 0.8	1.8 ± 1.2	1.7 ± 0.5	1.3 ± 0.2
DT MV [ms]	253 ± 151	130 ± 80	191 ± 77	213 ± 67
E/E’	7.1 ± 2.3	8.8 ± 3.5	5.6 ± 2.7	7.5 ± 2.8

Echocardiographic data of n = 7 swine that were resuscitated using a modified Impella ECP. The values are presented for baseline (BL), 30 (PR30), 300 (PR 300) and 360 (PR360) minutes following the return of spontaneous circulation. DT MV: deceleration time mitral valve, LV-EF: left ventricular ejection fraction, LV-GLS: left ventricular global longitudinal strain, RVD basal: right ventricular basal diameter, RV-GLS: right ventricular global longitudinal strain and TASV: tricuspid annular plane systolic velocity. The data are presented as the mean ± standard deviation (SD).

## Data Availability

The raw data generated and analyzed during the current study are available from the corresponding author upon reasonable request.

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
