# Peer review of "The Self-Expandable Impella CP (ECP) as a Mechanical Resuscitation Device"

_bioengineering, 2024, doi:10.3390/bioengineering11050456_

Round 1

Reviewer 1 Report

Comments and Suggestions for Authors

This study evaluates the feasibility of resuscitation using the Impella ECP in a swine cardiac arrest model. The methods and results are comprehensive. The following comments need to be addressed.

Comments:

1.     Introduction needs to be slightly extended.

2.     Figure 3 is the statistical results based figure. The authors should show an additional picture at least including some important hemodynamic parameters, such as aortic pressure, pulmonary artery pressure, and Impella flow rates.

3.     Table 2 should include results of Impella flow rates (L/min).

Reviewer 2 Report

Comments and Suggestions for Authors

A very promising option for improving the cardiopulmonary resuscitation complex. I would like to hear the authors' opinions on some issues. There is currently work on the use of ECMO for cardiopulmonary resuscitation, is there work on comparing the two technologies? Vasopressors are not considered standard of care for cardiac arrest. Does the use of vasopressors in your work justify the use of mechanical support only in the operating room or cath lab, but not in the setting of out-of-hospital cardiac arrest? How was neurological outcome assessed in animals? Has coronary blood flow been assessed in animals?
